# Identification of DNA methylation markers for age and Bovine Respiratory Disease in dairy cattle: A pilot study based on Reduced Representation Bisulfite Sequencing
E. Attree [1] ✉, B. Griffiths[2], K. Panchal[3], D. Xia [4], D. Werling[4], G. Banos[5], G. Oikonomou[2] & A. Psifidi [1,4] ✉

Methylation profiles of animals are known to differ by age and disease status. Bovine respiratory disease (BRD), a complex infectious disease, primarily affects calves and has significant impact on animal welfare and the cattle industry, due to production losses, increased veterinary costs as well as animal losses. BRD susceptibility is multifactorial, influenced by both environmental and genetic factors. We have performed a pilot study to investigate the epigenetic profile of BRD susceptibility in six calves (three healthy versus three diagnosed with BRD) and age-related methylation differences between healthy calves and adult dairy cows (three calves versus four adult cows) using Reduced Representation Bisulfite Sequencing (RRBS). We identified 2537 genes within differentially methylated regions between calves and adults. Functional analysis revealed enrichment of developmental pathways including cell fate commitment and tissue morphogenesis. Between healthy and BRD affected calves, 964 genes were identified within differentially methylated regions. Immune and vasculature regulatory pathways were enriched and key candidates in BRD susceptibility involved in complement cascade regulation, vasoconstriction and respiratory cilia structure and function were identified. Further studies with a greater sample size are needed to validate these findings and formulate integration into breeding programmes aiming to increase animal longevity and disease resistance.

Epigenetic changes are genetic modifications that influence gene regulation, without changing the DNA sequence. DNA methylation plays an important role in the development of disease[1,2] and ageing[3–6]. Although there are several relevant studies in humans and mice, there are limited studies in cattle. Differences in DNA methylation caused by environmental factors alter transcriptional regulation, for example by silencing genes, and thus, differential methylation is an important epigenetic modification that can greatly impact development and disease. DNA methylation changes are associated with age in humans and animals[7,8], these known age-related

changes can be used to construct epigenetic clocks. Epigenetic clocks are tools in the prediction of genetic age of tissues and individuals that can be used to predict chronological age or assist in identification of the influence of biological differences such as disease, individual genetics and environmental factors[4]. Epigenetic clocks have been constructed for human and mouse tissues, using methylation data from multiple tissue types[4,6]. Only recently have epigenetic clocks been constructed in cattle to predict chronological age[3,9], specifically for the prediction of oocyst epigenetic age and therefore reproductive aging, an important consideration in the dairy industry and

[1]Department of Clinical Science and Services, The Royal Veterinary College, Hatfield, UK. [2]Department of Livestock and One Health, Institute of Infection, Veterinary and Ecological Sciences, University of Liverpool, Neston, UK. [3]Institute of Applied Sciences, Charotar University of Science and Technology (CHAR-USAT), Gujarat, India. [4]Department of Pathobiology and Population Sciences, Centre for Vaccinology and Regenerative Medicine, Royal Veterinary College, Hatfield, UK. [5]Scotland's Rural College (SRUC), Easter Bush, Midlothian, Scotland, UK. ✉e-mail: eattree18@rvc.ac.uk; apsifidi@rvc.ac.uk

potentially a useful model for human reproductive aging[10]. Moreover, the role of DNA methylation in dairy cattle mastitis caused by *Staphylococcus aureus*[11] and *Escherichia coli*[12] has been investigated.

In addition to mastitis, early life calf losses cause a significant economic drain on the dairy industry as a dairy cow does not generate direct income until after its first calving[13]. The cost of rearing a replacement dairy cow has previously been estimated as 15–20% of the animal's economic production value[14]. Those heifers that die early or have to be culled before the end of their first lactation do not reimburse the economic cost of their rearing. Early life diseases have a significant impact on animal welfare, and thus on the cattle industry as a whole. Losses are typically attributed to impaired growth and consequential reduction in carcass value, the cost of disease management and treatment and, in severe cases, mortality[15–18].

Excluding abortions, still-births and calves that die within the first 24 hours after delivery, the incidence of neonatal mortality (calves aged from 1–28 days) has been reported to account for up to 12% of early calf losses[19,20]. The two most prevalent diseases contributing to this are diarrhoea and bovine respiratory disease complex (BRDC)[21], with clinical signs observed in pre-weaned calves at a prevalence of 13.8–21.6% and 8.1–22.8% respectively[22–24].

Multiple pathogens have been reported to contribute to the development of BRDC, including bacteria such as *Histophilus somni*, *Mannheimia hemolytica*, *Mycoplasma bovis* and *Pasteurella multocida*[25–30] and viruses such as bovine coronavirus (BoCoV), bovine viral diarrhoea virus (BVDV), bovine herpesvirus-1 (BHV-1), bovine reovirus, bovine respiratory syncytial virus (BRSV) and parainfluenzavirus-3 (BoPi-3)[18,31–34]. New born animals are at increased risk of infection with these pathogens due to their not yet fully developed adaptive immune system, environmental stressors (mainly weaning, transportation and contact with animals from other source), all of which increase potentially immunosuppression[35]. Multivalent vaccines against some of the causative agents of BRD are commercially available; these can be given systemically after the decline of maternally-derived antibodies (from 3 weeks of age onwards). However, these vaccines do not provide full protective immunity and there are further questions over the most efficient time of administration, intra-nasal at day 1 or systemically at week 3, to achieve maximum efficacy and the impact of management and stressors on efficiency[36,37]. Management of BRDC is typically by either curative of prophylactic administration of antibiotics[18,38]. This aids potentially the development of antimicrobial resistance[18] and therefore poses a risk to both, animal and human health.

An animal's response to complex diseases such as BRD is multifactorial, influenced by both the environment, the pathogen and individual genetic profiles. Previous studies have investigated the heritability of resistance to BRDC with estimates ranging from 0.07 to 0.29[39–41]. Moreover, it has been suggested that an epigenetic component may be involved since disease episodes early in a calf's life has been reported to negatively impact lifetime performance and age at first calving[42,43]. Therefore, studying the methylation profile of these calves may provide further data for future breeding programs designed to improve herd resistance[39,40,44,45] as well as increase our understanding of the underlying mechanisms of disease susceptibility. This endeavour would be greatly advantageous in improving genetic resistance to BRD infection, improving welfare of animals and subsequently reducing resultant economic losses. Further, reducing the instance of antibiotic treatment of non-affected animals would consequently reduce the risk of the development of antimicrobial drug resistance.

In the present pilot study we examined differential methylation between healthy calves and cows to establish baseline age related differential methylation for the purpose of identification of potential markers of genetic age. We also examined differences in methylation with respect to health status of calves to identify differential methylation between healthy and BRD affected that may convey resistance to disease for the purpose of identifying potential targets for genetic improvement of dairy cattle.

## Results
The animals involved in this study were all from one commercial dairy farm in North Wales and therefore raised under the same management. The four adult cows enrolled were female pedigree Holsteins aged 5-6 years old. Blood samples were collected as part of routine herd health screening of freshly calved cows, all in the same lactation stage. Blood samples, for DNA methylation analysis by reduced representation bisulfite sequencing (RRBS), were collected during routine screening after calving and animals were clinically healthy at the time of sampling. In addition, fourteen Holstein calves were enrolled into the study within one week of birth. The health of the 14 calves was assessed over an eight-week period from birth with Wisconsin scores[46] recorded at weeks one, five and eight (Table 1). Calves were ranked best to worst based on total score. Six of these calves were selected to compare their blood methylation profiles, three calves with the lowest total score were taken forward as the 'healthy' samples and three calves with some of the worst scores (scores 10–12 out of 14) were taken forward as the 'diseased' calf samples. Genetic relatedness of the calves was considered during selection: selected cases and controls were half siblings with calves 3 and 12 being twins with different BRD diagnoses (Supplementary table S1). The number of times respiratory disease was observed and the score was low and the same between calves 3 and 10, however, due to the higher lung scanning score of calf 10 it was designated to the disease group.

## Comparison of methylation profiles between healthy calves and cows
Firstly, the methylation profiles of healthy female calves ($n = 3$) and cows ($n = 4$) were compared using RRBS to form a baseline understanding of methylation differences based on animal age. Clear clustering and separation between the methylation profiles of cows and calves was observed (Fig. 1A, B). Between adults and calves 5256 (1.38%) bases were hypermethylated and 7100 (1.86%) bases were hypomethylated.

A total of 4794 regions were differentially methylated (≥25% difference of methylation between groups, $p \leq 0.05$ and $q \leq 0.01$) at the time of sampling between cow and calf samples, corresponding to 2537 genes (1995 protein coding, 31 pseudogenes, 51 rRNA, 205 lncRNA, 78 miRNA, 95 snRNA and 46 snoRNA). The distribution of differential methylation after annotation of gene features indicated that 6% of differential methylation was identified in promoter regions, 10% in exonic, 31% in intronic and 53% in intergenic regions.

Functional annotation enrichment and clustering analysis by Database for Annotation, Visualization and Integrated Discovery (DAVID)[47,48], Panther Over-Representation test[49] and Reactome pathway analysis[50] were conducted for all genes identified with statistically differentially methylated regions, to provide further insights on the underlying pathways and biological processes related to age. Functional enrichment analysis did not reveal overrepresentation for single gene ontology (GO) biological process terms. Nevertheless, DAVID functional annotation clustering analysis revealed 51 clusters with enrichment scores >1 (detailed in Supplementary Data 1), of which five clusters had enrichment scores ≥3, including: peptidyl-serine

**Table 1 | Wisconsin scores for weeks 1, 5 and 8 of the study and the total score over the eight weeks, the number of times respiratory disease was observed over the eight-week period and for each calf the lung scanning score as measured at week 8**

| Calf | Week 1 Wisconsin score total | Week 5 Wisconsin score total | Week 8 Wisconsin score total | Respiratory Disease Frequency[a] | Lung Scanning Score |
|---|---|---|---|---|---|
| 1 | 0 | 0 | 0 | 0 | 0 |
| 2 | 0 | 0 | 0 | 0 | 0 |
| 3 | 1 | 0 | 1 | 0 | 1 |
| 10 | 0 | 1 | 1 | 1 | 2 |
| 11 | 0 | 3 | 3 | 1 | 0 |
| 12 | 4 | 0 | 3 | 1 | 2 |

Calves 1–3 were considered 'healthy' and calves 10–12 'diseased'
[a]The number of times respiratory disease was detected from the week 1, 5 and 8 checks.

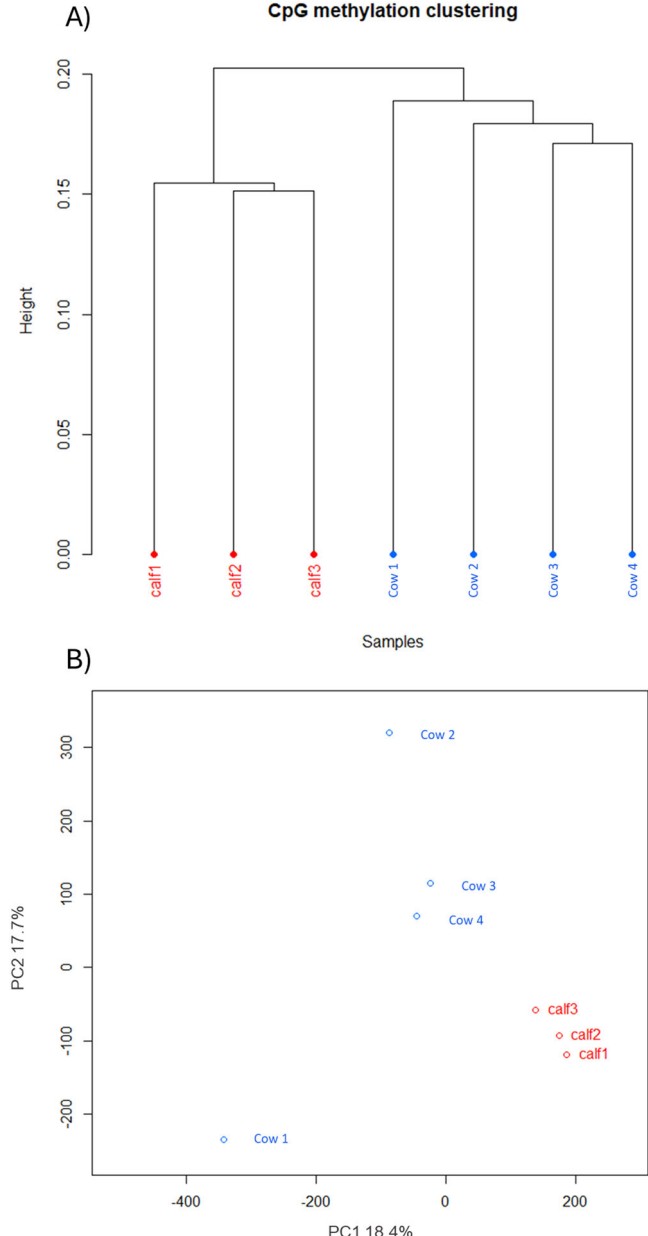

**Fig. 1 | Graphical representation of the CpG methylation clustering of healthy calves and adult cows.** **A** dendrogram representation of the CpG methylation clustering of healthy calves and adult cows. **B** PCA analysis indicating the clustering and separation of cows and calves.

phosphorylation (enrichment score 11), Src homology-3 domain (enrichment score 4.01), regulation of transcription from RNA polymerase II promoter (enrichment score 3.39), SH2 domain (enrichment score 3.12), and DNA-templated transcription initiation (enrichment score 3.03).

The genes identified within differentially methylated regions were also analysed by Panther overrepresentation test. A total of 123 GO biological process terms were found to be statistically significantly enriched ($p ≤ 0.0005$, FDR $≤ 0.05$), 113 over-represented and 10 under-represented, with fold enrichment ranging from 10.13 to 0.47 (supplementary Table S2). Enriched terms were primarily associated with developmental process, regulation of primary metabolic process, cell differentiation and anatomical structure development. Moreover, Reactome pathway analysis identified 35 significantly enriched pathways ($p ≤ 0.05$ and FDR $≤ 0.05$) involved in inflammation, signal transduction and the innate immune system, detailed in supplementary Table S3.

Age related methylation differences have previously been reported positionally within 50 kb of the transcription start site (TSS)[51], this threshold has also been previously applied to define *cis* regions of differential methylation[52]. Functional analysis of identified differential methylation was, therefore, restricted to 50 kb either side of the TSS, resulting in 1932 genes included for further analysis. The functional annotation clustering analysis with DAVID identified 36 clusters with enrichment scores >1 (detailed in Supplementary Data 1) with four of those having enrichment score >3: DNA-templated transcription, initiation (enrichment score 3.67), peptidyl-serine phosphorylation (enrichment score 3.65), Src homology-3 domain (enrichment score 3.27) and Pleckstrin homology domain (enrichment score 3.05). A total of 41 Reactome pathways were identified as significantly enriched ($p ≤ 0.01$, FDR $≤ 0.05$) with fold enrichment ranging from 1.47 to 8.05, detailed in supplementary Table S4, with the greatest enrichment identified in Regulation of RUNX1 Expression and Activity (fold enrichment 8.05), Synthesis of Leukotrienes (LT) and Eoxins (EX) (fold enrichment 6.71) and Carnitine metabolism (fold enrichment 5.75).

GO enrichment analysis of these genes by PANTHER Over-representation Test identified 38 over and under enriched terms ($p ≤ 0.0001$, FDR $≤ 0.05$), with the greatest enrichment observed in hexose import across plasma membrane and carbohydrate import across plasma membrane (fold enrichment >12.0) (Table 2 and Fig. 2). GO terms from the Panther over-representation test were further simplified and visualised using REVIGO to identify the key enriched pathways, presented in Fig. 3. We found enrichment for cell communication regulation of cellular process, metabolic process, cellular component organisation and the ERK1/2 cascade (involved in cell proliferation, differentiation, apoptosis and stress responses[53,54]).

Further stringent restriction of region of methylation with respect to the TSS to 10 kb either side was applied for further functional enrichment analysis. Two statistically significantly enriched GO biological process term were identified by DAVID functional enrichment analysis; DNA-templated transcription, initiation (GO:0006352) (fold enrichment: 7.8, q $≤ 0.0005$) and regulation of GTPase activity (GO:0043087) (fold enrichment: 5.2, q $≤ 0.05$). A total of 23 functional annotation clusters were identified with an enrichment score >1 (detailed in Supplementary Data 1), two with two clusters having an enrichment score >3: DNA-templated transcription, initiation (enrichment score 4.8) and negative regulation of endopeptidase activity (enrichment score 3.1). Moreover, Panther overrepresentation test found six statistically significant over-represented GO biological process terms, detailed in Table 3.

A protein interaction network analysis of the genes with differential methylation identified within 10 kb either side of the TSS was generated using STRING (version 11.5)[55], presented in Fig. 4. Clustering of interacting proteins into three clusters revealed pathways involved in AMPK signalling pathway, intracellular signal transduction, protein kinase, ATP binding transcriptional regulation and MAPK pathway (Fig. 4).

Extension of differential methylation analysis to base level identification of differential methylation in promoter and exonic regions identified 149 genes with differential methylation in their promoter regions and 229 in their exonic regions. No functional enrichment was, however, observed for GO biological process' by Panther overrepresentation test or Reactome pathways analysis.

## Comparison of methylation profiles between healthy calves and calves diagnosed with BRD

The methylation profiles of healthy and diseased calves, with respect to BRD, were compared in order to identify methylation differences that may be related to resistance to BRD. The analysis of differential methylation between healthy and diseased animals, revealed 1541 (0.58%) regions were hypermethylated and 1511 (0.57%) regions were hypomethylated. A total of 1029 regions were differentially methylated (≥25% difference of methylation between groups, $p ≤ 0.05$ and q $≤ 0.01$) between healthy and diseased samples corresponding to 964 genes. The distribution of differential methylation after annotation of gene features indicated 5% of differential methylation was identified in promoter

**Table 2 | The list of over and under enriched gene ontology (GO) biological process terms as identified by PANTHER Overrepresentation Test of all genes with differentially methylated regions within 50 kb of the TSS when healthy cows and calves compared**

| GO biological process | Fold enrichment | P value | FDR |
|---|---|---|---|
| hexose import across plasma membrane (GO:0140271) | 12.88 | 7.05E-05 | 2.96E-02 |
| carbohydrate import across plasma membrane (GO:0098704) | 12.88 | 7.05E-05 | 2.87E-02 |
| retinoic acid receptor signalling pathway (GO:0048384) | 6.63 | 3.94E-05 | 2.30E-02 |
| negative regulation of ERK1 and ERK2 cascade (GO:0070373) | 4.43 | 2.22E-05 | 2.48E-02 |
| regulation of hydrolase activity (GO:0051336) | 1.88 | 5.39E-05 | 2.59E-02 |
| regulation of catalytic activity (GO:0050790) | 1.66 | 4.24E-05 | 2.19E-02 |
| phosphorylation (GO:0016310) | 1.64 | 1.28E-04 | 4.40E-02 |
| positive regulation of cell communication (GO:0010647) | 1.47 | 4.61E-05 | 2.29E-02 |
| positive regulation of signalling (GO:0023056) | 1.46 | 6.09E-05 | 2.64E-02 |
| negative regulation of nitrogen compound metabolic process (GO:0051172) | 1.42 | 3.07E-05 | 2.29E-02 |
| negative regulation of macromolecule biosynthetic process (GO:0010558) | 1.4 | 8.43E-05 | 3.15E-02 |
| regulation of signal transduction (GO:0009966) | 1.4 | 1.76E-06 | 3.93E-03 |
| regulation of cell communication (GO:0010646) | 1.39 | 4.86E-07 | 2.17E-03 |
| negative regulation of cellular biosynthetic process (GO:0031327) | 1.39 | 1.23E-04 | 4.35E-02 |
| regulation of developmental process (GO:0050793) | 1.38 | 8.57E-05 | 3.11E-02 |
| negative regulation of cellular metabolic process (GO:0031324) | 1.38 | 3.66E-05 | 2.34E-02 |
| regulation of multicellular organismal process (GO:0051239) | 1.38 | 1.39E-05 | 2.08E-02 |
| negative regulation of macromolecule metabolic process (GO:0010605) | 1.37 | 2.54E-05 | 2.63E-02 |
| regulation of signalling (GO:0023051) | 1.37 | 1.51E-06 | 5.07E-03 |
| negative regulation of metabolic process (GO:0009892) | 1.35 | 3.60E-05 | 2.42E-02 |
| regulation of response to stimulus (GO:0048583) | 1.34 | 1.62E-06 | 4.34E-03 |
| cellular component organization (GO:0016043) | 1.3 | 2.91E-09 | 3.91E-05 |
| cellular component organization or biogenesis (GO:0071840) | 1.27 | 5.60E-08 | 3.76E-04 |
| negative regulation of cellular process (GO:0048523) | 1.26 | 2.57E-05 | 2.16E-02 |
| negative regulation of biological process (GO:0048519) | 1.25 | 2.56E-05 | 2.29E-02 |
| positive regulation of cellular process (GO:0048522) | 1.23 | 1.81E-05 | 2.43E-02 |
| positive regulation of biological process (GO:0048518) | 1.22 | 2.55E-05 | 2.44E-02 |
| regulation of macromolecule metabolic process (GO:0060255) | 1.21 | 2.67E-05 | 2.11E-02 |
| regulation of metabolic process (GO:0019222) | 1.2 | 3.23E-05 | 2.28E-02 |
| biological regulation (GO:0065007) | 1.11 | 3.96E-05 | 2.22E-02 |
| biological_process (GO:0008150) | 1.05 | 5.44E-05 | 2.52E-02 |
| sensory perception (GO:0007600) | 0.59 | 1.90E-05 | 2.32E-02 |
| detection of stimulus involved in sensory perception (GO:0050906) | 0.57 | 7.88E-05 | 3.03E-02 |
| detection of chemical stimulus involved in sensory perception (GO:0050907) | 0.57 | 7.67E-05 | 3.03E-02 |
| sensory perception of chemical stimulus (GO:0007606) | 0.56 | 4.11E-05 | 2.21E-02 |
| detection of chemical stimulus (GO:0009593) | 0.56 | 3.93E-05 | 2.40E-02 |
| detection of chemical stimulus involved in sensory perception of smell (GO:0050911) | 0.51 | 7.38E-06 | 1.24E-02 |
| sensory perception of smell (GO:0007608) | 0.50 | 4.53E-06 | 8.69E-03 |

regions, 9% in exonic regions, 30% in intronic regions and 56% in intergenic regions.

Similarly with the age comparison, functional annotation enrichment and clustering analysis by DAVID[47,48], Panther Over-Representation test[49] and Reactome pathway analysis[50] were conducted for all genes identified with statistically differentially methylated regions, to provide further insights on the underlying pathways and biological processes related to BRD resistance. Although in the functional annotation enrichment analysis none of the terms remained statistically significant after correcting for multiple testing, for several biological processes a high fold change without statistical significance was identified. Specifically, the greatest fold enrichment of biological process was observed for: regulation of centrosome cycle (GO:0046605, fold change 19.51), positive regulation of renal sodium excretion (GO:0035815, fold change 19.51), positive regulation of plasma cell differentiation (GO:1900100, fold change 19.51), zygotic specification of dorsal/ventral axis (GO:0007352, fold change 19.51), negative regulation of non-canonical Wnt signalling pathway (GO:2000051, fold change 14.63) and positive regulation of T-helper 2 cell cytokine production (GO:2000553, fold change 11.15).

Functional annotation clustering analysis using DAVID revealed 24 clusters with enrichment scores >1 (detailed in Supplementary Data 1), those with the highest scores included: chloride transmembrane transport (enrichment score 2.58), homophilic cell adhesion via plasma membrane adhesion molecules (enrichment score 2.46) and Fibronectin type-III (enrichment score 2.42).

Panther Overrepresentation test of these genes identified 24 statistically significant over (21) and under (3) enriched GO biological process terms

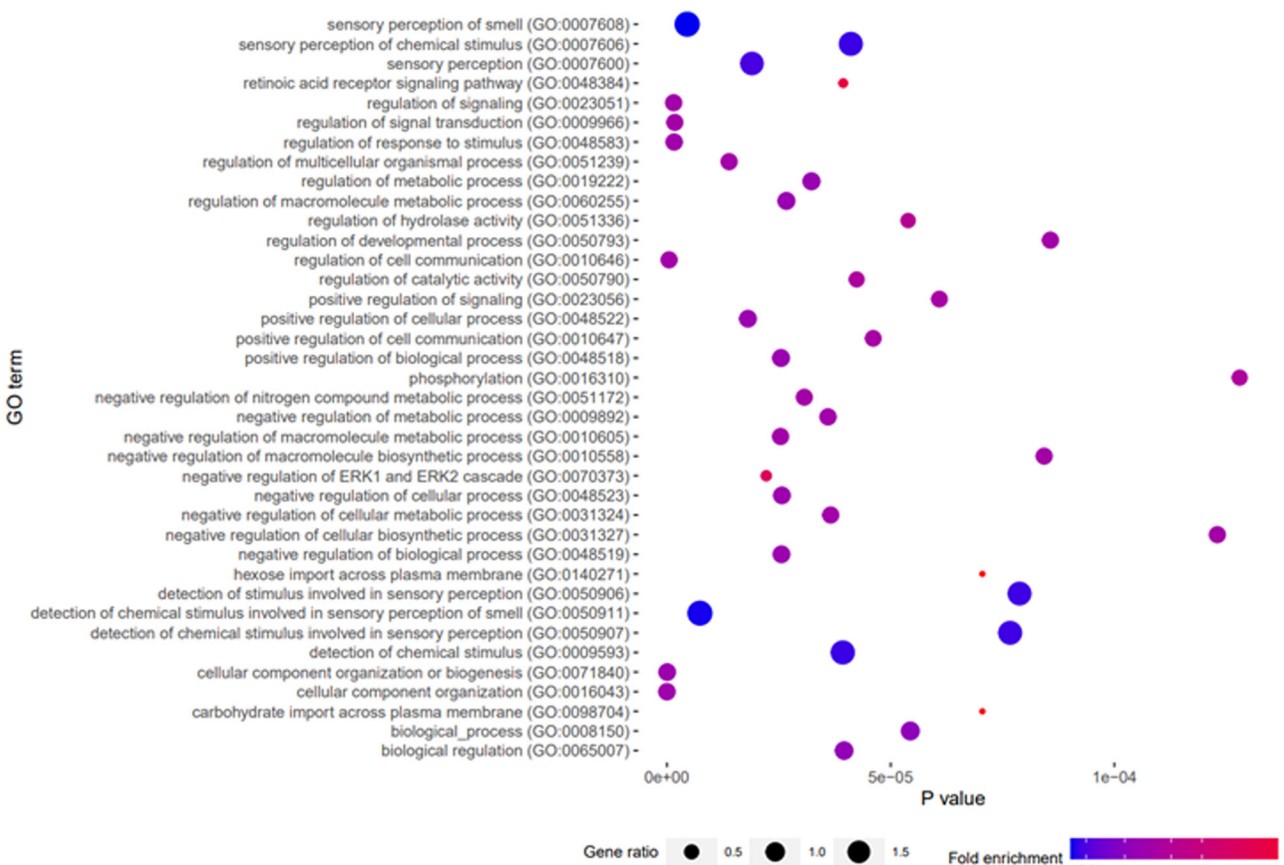

**Fig. 2 | Enriched GO biological process' of differentially methylated regions between healthy cows and calves.** Graphical representation of over and under-represented GO biological process' identified by PANTHER Overrepresentation Test of the genes identified within 50 kb of differentially methylated regions identified between healthy cows (as control) and calves. Gene ratio was calculated by gene count per term / number of genes assigned the term in the genome.

($p \leq 0.05$, FDR $\leq 0.05$). (Table 4). Functional pathway analysis by Reactome of all genes associated with differentially methylated regions identified three statistically significant over-represented pathways, including: neuronal system (fold enrichment 2.29, $p$-value 8.19E-06, FDR 5.42E-03), adaptive immune system (fold enrichment 1.79, $p$-value 4.55E-05, FDR 1.51E-02) and signal transduction (fold enrichment 1.4, $p$-value 3.94E-05, FDR 1.74E-02).

A 50 kb threshold for methylated regions either side of the TSS was applied as a previously defined *cis* region[52] and previously associated with response to bacterial infection in humans[56]. A total of 2553 genes with differentially methylated regions within 50 kb (either side) of the TSS were taken forward for functional analysis. Functional enrichment analysis by DAVID did not reveal statistically significant enrichment for single GO terms after correcting for multiple testing. Nevertheless, several relevant terms with a nominal significance and a fold change (>6) were identified. Specifically, these included: aortic valve morphogenesis (GO:0003180), positive regulation of chondrocyte differentiation (GO:0032332), positive regulation of calcineurin-NFAT signalling cascade (GO:0070886), negative regulation of vascular permeability (GO:0043116), positive regulation of T-helper 2 cell cytokine production (GO:2000553).

GO terms from the DAVID functional enrichment analyses were further simplified and visualised using REVIGO to identify the key enriched pathways (Fig. 5).Those of particular relevance to BRD that were identified included those involved in epithelial cell regulation and vascular permeability and immune regulation, for example: positive regulation of T-helper 2 cell cytokine production (lymphocytes involved in the adaptive immune response recruiting other activated immune cells and the production of antibodies[57]), negative regulation of vascular permeability, angiogenesis and cell adhesion (Fig. 5).

Functional annotation clustering analysis using DAVID revealed the presence of 17 clusters with enrichment scores >1 (detailed in Supplementary Data 1). The greatest enrichment was identified for the clusters Calpain cysteine protease (enrichment score 2.22), endoplasmic reticulum unfolded protein response (enrichment score 1.79), BMERB domain (enrichment score 1.54) and Fibronectin type-III (enrichment score 1.78),calcium-dependent cysteine-type endopeptidase activity (enrichment score 2), Fibronectin, type III (enrichment score 1.78) and dilute domain (enrichment score 1.78).

Panther Overrepresentation test found nine underrepresented GO terms with statistical significance ($p \leq 0.05$, FDR $\leq 0.05$), no over-representation was observed. These GO terms were primarily involved in metabolic processes: regulation of biological process (GO:0050789), metabolic process (GO:0008152), cellular process (GO:0009987), biological regulation (GO:0065007), biological_process (GO:0008150), organic substance metabolic process (GO:0071704), nitrogen compound metabolic process (GO:0006807), primary metabolic process (GO:0044238) and cellular nitrogen compound metabolic process (GO:0034641).

More stringent restriction of region of methylation with respect to the TSS to 10 kb either side was applied for further functional enrichment analysis. No single GO biological process was found to be statistically significantly enriched by DAVID functional enrichment analysis after correction for multiple testing, however, the greatest fold enrichment was observed for the GO biological process terms detailed in Table 5. Functional annotation clustering analysis revealed seven clusters with enrichment scores >1 (detailed in Supplementary Data 1). The greatest enrichment was observed for: BMERB domain (enrichment score 2.19), FAD binding (enrichment score 1.99), Ras-associating domain (enrichment score 1.57)

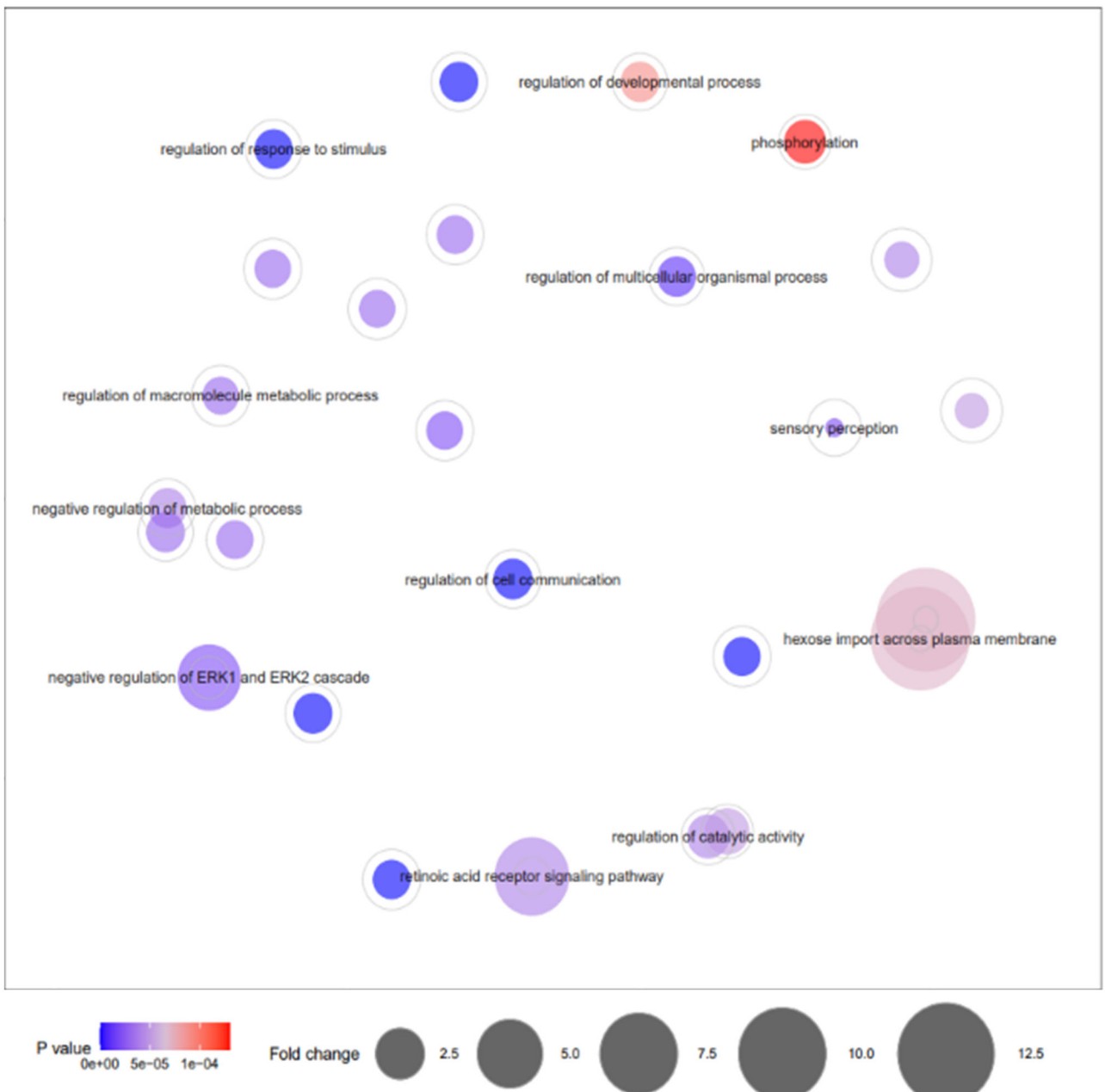

**Fig. 3 | REVIGO representation of the enriched GO biological process' of differentially methylated regions between healthy cows and calves.** Graphical visualisation by REVIGO of over and under-represented GO biological process' identified by Panther overrepresentation test of the genes identified within 50 kb of differentially methylated regions identified between healthy cows and calves.

**Table 3 | Statistically significant (FDR ≤ 0.05) over-represented GO biological process terms of the identified genes with differential methylation within 10 kb of the transcription start site based on Panther overrepresentation test**

| GO accession | GO term | Fold change |
| --- | --- | --- |
| GO:0051647 | nucleus localization | 7.80 |
| GO:0051336 | regulation of hydrolase activity | 2.57 |
| GO:0050790 | regulation of catalytic activity | 2.19 |
| GO:0065009 | regulation of molecular function | 1.87 |
| GO:0016043 | cellular component organization | 1.36 |
| GO:0071840 | cellular component organization or biogenesis | 1.31 |

and endosome (enrichment score 1.19); Immunoglobulin domain enrichment score 1.05.

A protein interaction network analysis of genes with differential methylation identified within 10 kb of a TSS was generated using STRING (version 11.5)[55], presented in Fig. 6. Protein interaction networks identified enrichment of pathways involved in the mitochondrial membrane respiratory chain, Cytochrome P450 pathway, ribosomal activity, Ser/Thr protein kinase, fibroblast growth factor receptor and regulation of endothelial cell migration and sprouting angiogenesis.

Extension of differential methylation analysis to base level identification of differential methylation in promoter and exonic regions identified 4763 genes with differential methylation in their promoter regions and 65 in their exonic regions. Panther overrepresentation test revealed 23 statistically significant enriched GO biological process' in promoter regions, shown in

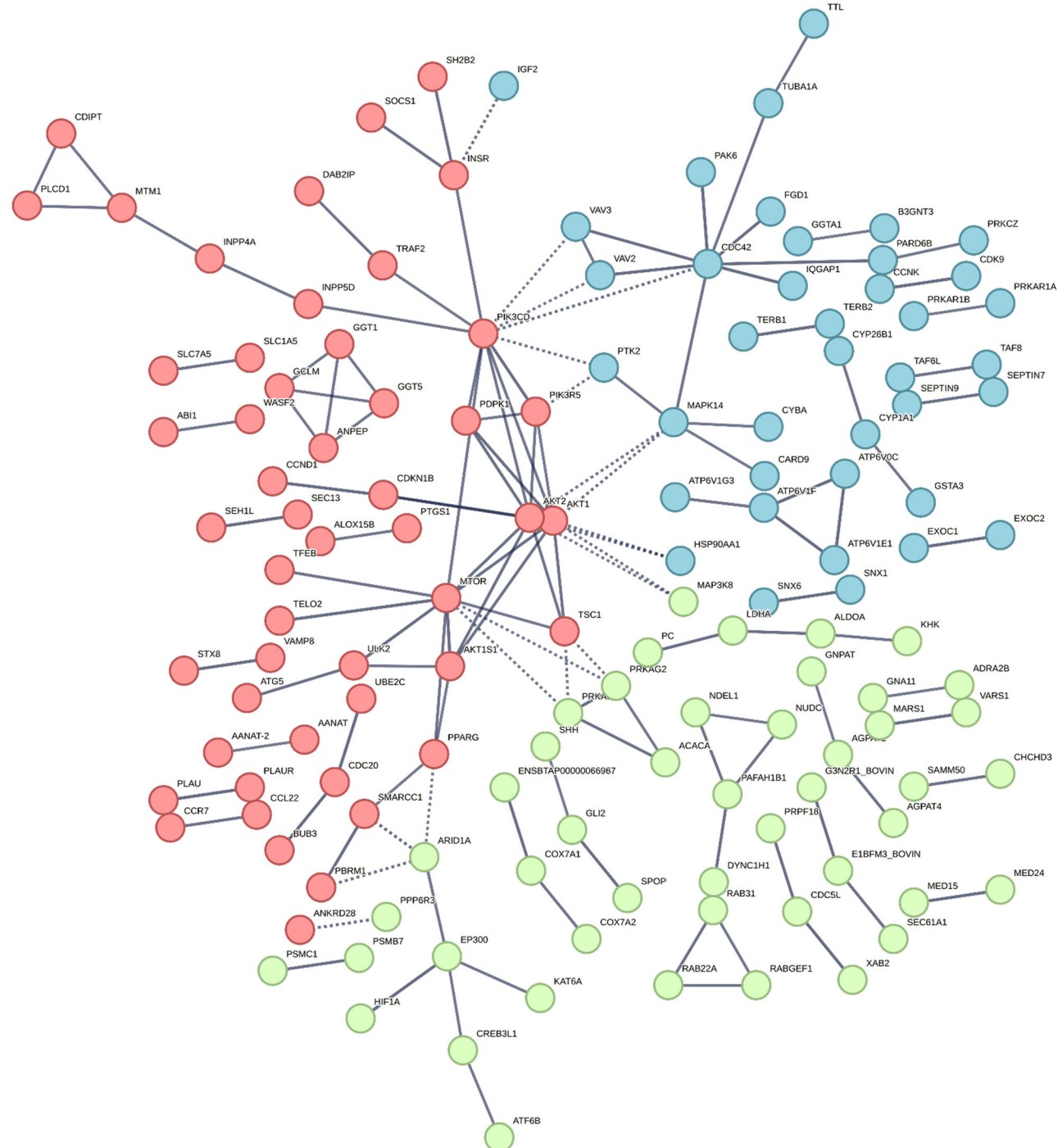

**Fig. 4 | Protein interaction network of differentially methylated genes between healthy cows and calves.** String protein interaction network of identified genes with differential methylation between healthy cows and calves within 10 kb of a TSS. The network was partitioned into three clusters using the k-means algorithm and color-coded as red, blue, and green.

Table 6, for example: response to stimulus and cellular process. Additionally, three statistically significant enriched Reactome pathways: protein-protein interactions at synapses (fold enrichment = 2.86, $p$ = 4.77E-05, FDR = 2.04 E-02), neuronal system (fold enrichment =2.01, $p$ = 1.21E-08, FDR = 6.87E-06) and signal transduction (fold enrichment =1.35, $p$ = 4.19E-09, FDR = 3.58E-06) were identified. No functional enrichment was observed for genes with differential methylation in exonic regions.

## Discussion
We set out this pilot study to identify the differences in methylation profiles of healthy adults and calves and between healthy calves and those diagnosed

with BRD using RRBS for the purpose of identification of markers of genetic age and of potential targets for genetic improvement of dairy cattle. Methylation differences between healthy calves and cows were primarily in regions associated with genes involved in growth and developmental pathways, as to be expected. Using GO and pathway enrichment analyses we identified enrichment of cell organisation, metabolism, signalling and sensory perception. Pathways identified consistently as enriched across the age comparison included the MAPK and ERK pathways, which have previously been associated with ageing[58,59]. Between healthy and diseased calves, the identified differences in methylation were primarily in regions associated with genes involved in immune regulation and pulmonary pathways. From

**Table 4 | List of over and under enriched GO biological process terms as identified by PANTHER Overrepresentation Test of all genes identified with differentially methylated regions between healthy calves and those with BRD**

| GO biological process complete | fold Enrichment | P value | FDR |
|---|---|---|---|
| cell junction assembly (GO:0034329) | 2.54 | 7.7E-05 | 4.6E-02 |
| cell junction organization (GO:0034330) | 2.18 | 1.6E-05 | 1.9E-02 |
| neuron development (GO:0048666) | 1.94 | 3.6E-06 | 6.5E-03 |
| cell morphogenesis (GO:0000902) | 1.89 | 7.1E-05 | 4.6E-02 |
| neuron differentiation (GO:0030182) | 1.83 | 4.2E-06 | 6.7E-03 |
| generation of neurons (GO:0048699) | 1.82 | 2.1E-06 | 4.9E-03 |
| neurogenesis (GO:0022008) | 1.75 | 2.8E-06 | 5.7E-03 |
| plasma membrane bounded cell projection organization (GO:0120036) | 1.69 | 2.6E-05 | 2.5E-02 |
| cell projection organization (GO:0030030) | 1.67 | 3.3E-05 | 2.6E-02 |
| nervous system development (GO:0007399) | 1.58 | 4.7E-06 | 6.8E-03 |
| system development (GO:0048731) | 1.51 | 6.4E-08 | 9.1E-04 |
| animal organ development (GO:0048513) | 1.50 | 1.0E-06 | 3.7E-03 |
| cell development (GO:0048468) | 1.48 | 4.4E-05 | 3.0E-02 |
| anatomical structure morphogenesis (GO:0009653) | 1.48 | 3.5E-05 | 2.6E-02 |
| multicellular organism development (GO:0007275) | 1.46 | 9.1E-08 | 6.5E-04 |
| negative regulation of cellular metabolic process (GO:0031324) | 1.45 | 7.6E-05 | 4.7E-02 |
| cellular developmental process (GO:0048869) | 1.38 | 1.9E-05 | 2.1E-02 |
| cell differentiation (GO:0030154) | 1.38 | 2.7E-05 | 2.3E-02 |
| anatomical structure development (GO:0048856) | 1.36 | 6.1E-07 | 2.9E-03 |
| developmental process (GO:0032502) | 1.33 | 1.1E-06 | 3.2E-03 |
| cellular component organization (GO:0016043) | 1.27 | 1.3E-05 | 1.6E-02 |
| detection of stimulus (GO:0051606) | 0.48 | 3.5E-05 | 2.5E-02 |
| detection of stimulus involved in sensory perception (GO:0050906) | 0.46 | 2.6E-05 | 2.4E-02 |
| detection of chemical stimulus involved in sensory perception of smell (GO:0050911) | 0.43 | 2.4E-05 | 2.4E-02 |

the GO and pathway enrichment analyses used in this work enrichment was identified in detection of stimulus, T-helper cell cytokine production, negative regulation of vascular permeability and angiogenesis. These identified genes and regions potentially play a role in susceptibility to BRD, however, this was a pilot study with a small sample size. We anticipate that a larger study will decipher further the role of DNA methylation in BRD susceptibility and age differences. Moreover, due to limited resources only a small number of calves were monitored resulting in the differences in respiratory health of the top and bottom three calves not being significantly different between all animals, calf 3 and 10 for example differ only in lung scanning score. We anticipate further differences in DNA methylation profile would have been identified in examination of calves with more significant differences in respiratory health score. Further validation of the results of this work would be necessary in order to be used to inform breeding programs to improve future herd health.

A higher percentage of hypomethylation observed in adult cows compared to calves, which is consistent with the established theory of genomic hypomethylation associated with ageing[60,61], the small percentage however is likely associated with the relatively young age of the adults (5-6 years old). Further the identification of differential methylation and enrichment of AKT Serine/Threonine Kinase 2 (*AKT2*) pathways between healthy cows and calves is an intriguing finding due to the implicated role of *AKT2* in aging. Murine experimental studies found *AKT2* ablation resulted in prolonged life-span with observed reduction in negative cardiac remodelling and a specific reduction in Ca2+ defects in contractile and intracellular cardiac tissue and mitochondrial injury, both typically associated with age[62]. Differential methylation affecting the regulation of this gene could therefore act as an important marker to establishing genetic ageing. It is, however, important to also note the role of *AKT2* in the immune system. *AKT2* is involved in macrophage polarization, regulation of the functions of dendritic cells and proliferation of T regulatory cells and *AKT2* ablation has

been shown to result in macrophage polarisation to M2 macrophages involved in wound healing[63]. Due to the young age of calves included in the comparison, the enrichment of this pathway could also be indicative of age-associated immune system development and maturation. Given that calves still have a relatively naïve adaptive immune system, the innate side plays a potentially more important role, and for this, a more pro-inflammatory M1 phenotype of macrophages is needed.

Previous studies of differential methylation in aging have led to the production of epigenetic clocks, for establishing an organisms biological age[8,64,65], and identification of age related biological process with differential methylation[66]. In beef cattle, age related hypermethylation was observed in promoter and 5' UTR regions, with hypomethylation in other regions. The observed hypermethylation was also found to be functionally linked to pathways including: DNA and transcription factor binding and regulation of genes associated with development and morphogenesis pathways and metabolism[65]. Similarly, previous functional enrichment analyses of hypermethylated genes in age comparisons of mice identified enrichment for functions such as system development, anatomical structure development, nervous system development, multicellular organismal development, cell fate commitment, cell differentiation, cell development, developmental process, cellular developmental process and multicellular organism process[66]. In our study GO enrichment analysis of genes with methylation differences between healthy cows and calves were primarily involved in developmental pathways, for example tissue morphogenesis, neuron differentiation, regulation of developmental process, cell fate commitment and organ morphogenesis, consistent with previous findings[65,66]. These enriched GO terms are expected as calves are still developing into adulthood, therefore, differential methylation of genes involved in these pathways could be useful candidates as markers of epigenetic ageing. Interestingly a reduction of GO enrichment of detection of chemical stimulus involved in sensory perception of smell was observed between healthy cows and calves

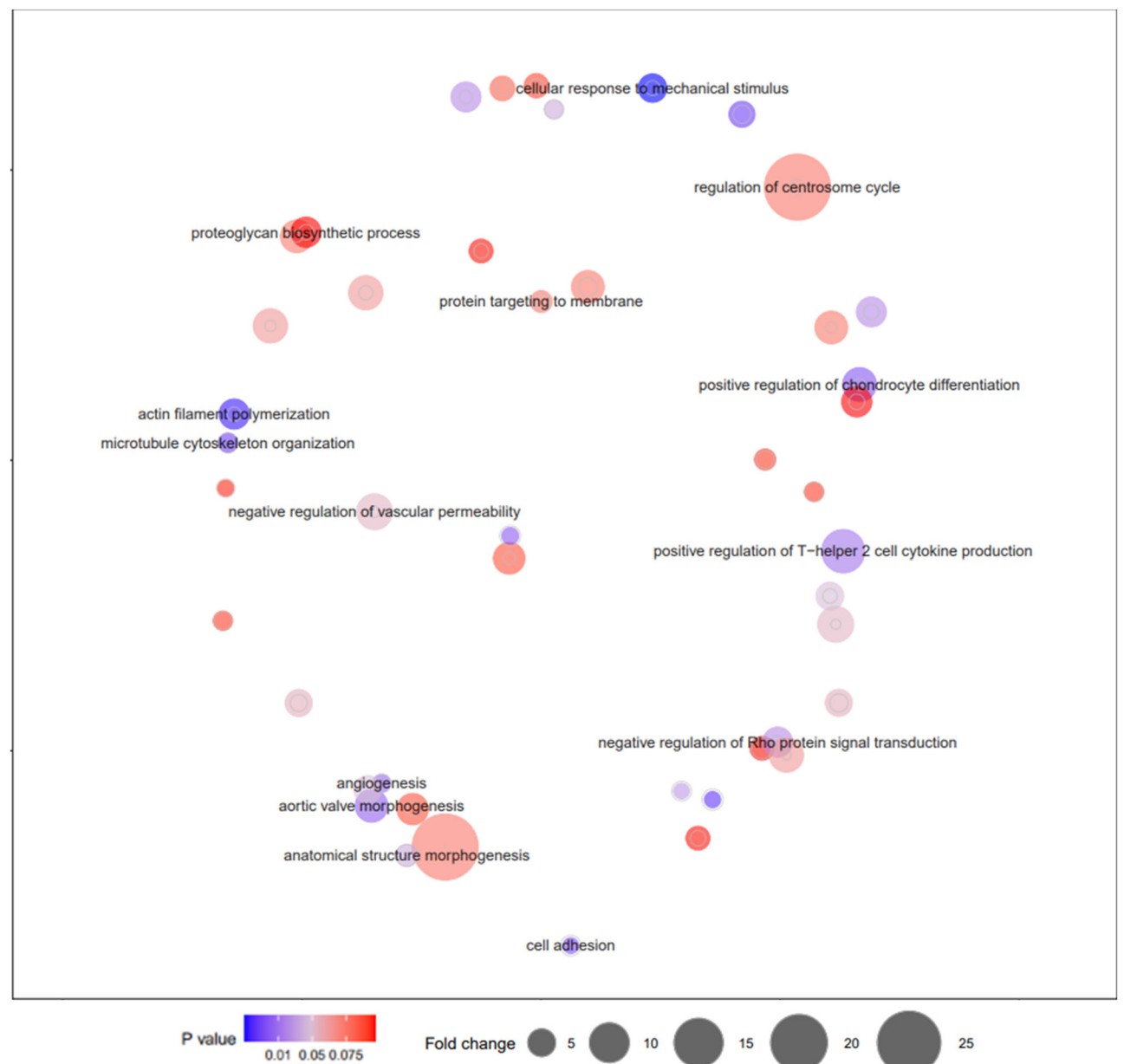

**Fig. 5 | REVIGO representation of the enriched GO biological process' of differentially methylated regions between healthy calves and those with BRD.** Graphical visualisation by REVIGO of over and under-represented GO biological process' identified by DAVID functional enrichment analysis of the genes within 50 kb of differentially methylated regions identified between healthy calves and those diagnosed with BRD.

**Table 5 | The GO biological process with the greatest fold enrichment as identified by DAVID functional enrichment analysis**

| GO accession | Term | Fold change |
|---|---|---|
| GO:0046605 | regulation of centrosome cycle | 38.89 |
| GO:0045026 | plasma membrane fusion | 38.89 |
| GO:0000738 | DNA catabolic process, exonucleolytic | 29.17 |
| GO:0061370 | testosterone biosynthetic process | 29.17 |
| GO:0032510 | endosome to lysosome transport via multivesicular body sorting pathway | 23.34 |
| GO:0032713 | negative regulation of interleukin-4 production | 23.34 |

and between healthy calves and those diagnosed with BRD. Many genes attributed in this GO annotation code for G protein coupled receptors, involved in cell communication and receptors to a range of stimuli, suggesting potentially enrichment of a more general signalling function outside of the olfactory system. The Reactome analysis identified enrichment for pathways related to inflammation, innate immune system and signal transduction which have all been previously reported to have an association with aging[67,68]. In cows specifically, age related hypomethylation of the thymus has been described[69] and in humans age related T-lymphocyte hypomethylation[70].

Resistance to BRD in dairy calves is a complex heritable trait[39,40,44], and therefore identification of genetic loci and candidate genes associated with susceptibility to BRD provide the opportunity for genomic selection aiming to reduce disease incidence. Nevertheless, epigenetic changes may also play a role since a significant number of genes have been identified in this study as

**Fig. 6 | Protein interaction network of differentially methylated genes between healthy calves and those with BRD.** String protein interaction network of identified genes with differential methylation between healthy and BRD calves within 10 kb of a TSS. The network was partitioned into three clusters using the k-means algorithm and color-coded as red, blue, and green.

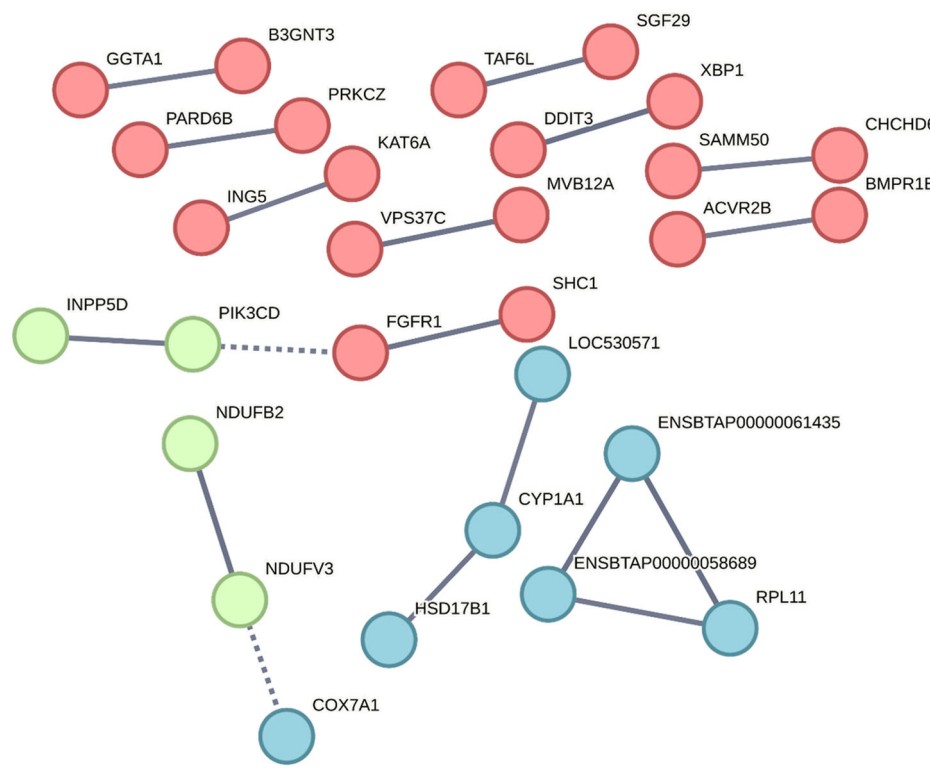

differentially methylated between healthy calves and those diagnosed with BRD. Although we have not generated RNA-Sequencing data from these calves to compare their blood transcriptomic profiles, when we compared our results with the results of previous transcriptomics, eQTL and genome-wide association (GWAS) studies[45,71] many of the differentially methylated genes have been also identified and prioritised as good candidate genes for BRD resistance in these studies[45,71]. Specifically, eight genes (*KPNA6, NEIL3, ADCK1, ZNF507, REEP3, AHSA1, SEPTIN11* and *IgLON5*) identified as differentially methylated were also identified within significant GWAS windows in the study of BRD susceptibility in pre-weaned Holstein calves[71]. Further, some of the differentially methylated genes (*TGM3, NRG1, RETN, IL1R2, ADGRE1, ALPL, KREMEN1, TRPC5, MN1, ALOX5AP, DYSF, ITGA9, NUPR1, SLC28A3, IGLON5, PTPN5, GLT1D1, DPYS, WIPI1, CFB, LOC511106, IL3RA*) also overlapped with previously identified differentially expressed genes in studies of BRD susceptibility of pre-weaned calves and feedlot cattle[45,71]. Of these, *NUPR1, CFB and ADGRE1* were also associated with identified *trans*-eQTLs for BRD resistance[45].

Moreover, our study of genes with differential methylation within 50 kb of the TSS revealed that inflammatory response, pulmonary epithelial integrity, host innate and adaptive immune system regulation were significantly enriched pathways, similar as to what was reported in two previously conducted enrichment analysis of genetic data (GSEA-SNP) and multi-omic (transcriptomic, genomic and metabolomic) studies of BRD resistance in crossbred or multi-breed beef cattle and Holstein calves[45,72]. Indeed, a number of innate immune related differentially methylated genes overlapped with existing identified important genes in BRD through genetic and transcriptomic studies; *PTPN5* (involved in interleukin-37 signalling), *ILR3A* (Interleukin 3 receptor subunit alpha), *IL1R2* (Interleukin 1 receptor type 2), *CFB* (involved in activation of C3 and C5 and alternative complement activation), *IGLON5* (immunoglobulin family protein), suggesting that epigenetic changes at the methylation level may underly these gene expression differences. The enrichment of the innate immune related genes in particular are consistently identified as important, in accordance with findings in pre-existing studies[45], likely because at the age of calves at the time of sampling the adaptive immune system would not have been fully matured, unlike the innate immune system.

A primary example of a good potential candidate gene identified in this work, with differential methylation in the promoter region, is dynein axonemal intermediate chain 1 (*DNAI1*), part of the dynein complex found in respiratory cilia with a function in humans in regulating dynein activity. The means in which the motion and therefore function of the cilia is achieved[73,74]. Moreover, genes with reported functions in inflammation pathways and in pulmonary vasculature are good candidates since in respiratory disease pulmonary vascular endothelial cells are among the primary cells to be damaged. Furthermore, platelet endothelial cell adhesion (*PECAM1*), also with differential methylation in the promoter region, is an adhesion molecule that facilitates leukocyte trans-endothelial migration during inflammation and is expressed on the surface of pulmonary vascular endothelial cells[75] and also interacts with the von Willebrand factor (*VWF*) which functions by promoting platelet adhesion to sites of vascular injury[76].

Identification of differential methylation at base level resolution applied to promoter and exonic regions revealed a significantly high number of bases as differentially methylated in the promoter region of the gene that encodes CUB domain containing protein 2 (*CDCP2*). CUB domain containing proteins are highly conserved with a diverse range of functions, for example: complement activation, inflammation, tissue repair and angiogenesis to name a few. In human studies, CUB domain containing protein 1 (*CDCP1*) expression has been found to strongly correlate with poor prognosis and relapse of lung adenocarcinoma patients[77,78]. The identification of the differentially methylated bases in this gene and its implication in human lung adenocarcinoma could be informative of a potential role in BRD. Significant hypomethylation in calves with BRD was also observed in the promoter region of dynein axonemal intermediate chain 1 (*DNAI1*) upon base level resolution analysis. A high number of bases were also identified as hypomethylated between healthy and BRD calves in exonic regions of the endothelin converting enzyme 1 (*ECE1*) gene. This gene functions by converting big endothelin to biologically active peptides with roles in vasoconstriction[79,80]. Increased expression of *ECE1* and its target endothelin-1 (*Et-1*) has previously been implicated in studies in pulmonary fibrosis in rat models[81–84] and in human idiopathic pulmonary fibrosis and correlated with disease activity[79,84]. This, in combination, with the identified enrichment of circulatory system development and morphogenesis of epithelium

**Table 6 | The enriched biological process' identified for the genes with differential methylation between healthy and BRD calves identified in their promoter regions as determined by PANTHER Overrepresentation Test Fisher Exact test**

| GO biological process | Fold Enrichment | *P*-value | FDR |
|---|---|---|---|
| biological regulation (GO:0065007) | 0.79 | 3.15E-06 | 2.27E-03 |
| regulation of biological process (GO:0050789) | 0.79 | 4.75E-06 | 3.02E-03 |
| response to stimulus (GO:0050896) | 0.76 | 1.81E-05 | 9.78E-03 |
| cellular process (GO:0009987) | 0.72 | 4.84E-15 | 1.75E-11 |
| biological_process (GO:0008150) | 0.7 | 2.25E-20 | 1.22E-16 |
| cellular metabolic process (GO:0044237) | 0.69 | 2.46E-08 | 2.96E-05 |
| organonitrogen compound metabolic process (GO:1901564) | 0.66 | 1.13E-07 | 1.11E-04 |
| protein metabolic process (GO:0019538) | 0.66 | 5.46E-06 | 3.28E-03 |
| primary metabolic process (GO:0044238) | 0.66 | 3.02E-11 | 5.44E-08 |
| metabolic process (GO:0008152) | 0.66 | 4.22E-13 | 1.14E-09 |
| organic substance metabolic process (GO:0071704) | 0.66 | 2.83E-12 | 6.12E-09 |
| nitrogen compound metabolic process (GO:0006807) | 0.65 | 7.15E-11 | 1.11E-07 |
| biosynthetic process (GO:0009058) | 0.65 | 2.25E-05 | 1.16E-02 |
| cellular biosynthetic process (GO:0044249) | 0.64 | 8.25E-05 | 3.72E-02 |
| organic substance biosynthetic process (GO:1901576) | 0.64 | 2.46E-05 | 1.21E-02 |
| organic cyclic compound metabolic process (GO:1901360) | 0.64 | 4.47E-06 | 3.02E-03 |
| macromolecule metabolic process (GO:0043170) | 0.64 | 4.78E-10 | 6.46E-07 |
| cellular aromatic compound metabolic process (GO:0006725) | 0.62 | 2.49E-06 | 1.92E-03 |
| cellular nitrogen compound metabolic process (GO:0034641) | 0.6 | 8.09E-08 | 8.76E-05 |
| heterocycle metabolic process (GO:0046483) | 0.59 | 5.38E-07 | 4.85E-04 |
| nucleobase-containing compound metabolic process (GO:0006139) | 0.59 | 1.13E-06 | 9.42E-04 |
| nucleic acid metabolic process (GO:0090304) | 0.59 | 1.29E-05 | 7.34E-03 |
| macromolecule biosynthetic process (GO:0009059) | 0.56 | 7.62E-05 | 3.58E-02 |

is consistent with existing identified enrichment of pulmonary epithelial integrity[72] and is suggestive of an important mechanism in susceptibility to BRD.

## Conclusions

In conclusion, functional enrichment of differentially methylated genes between healthy adult cows and healthy calves identified the expected developmental pathways as well as pathways involved in inflammation, the innate immune system and signal transduction. The observed differential methylation of inflammation, the innate immune system and signal transduction pathways are consistent with existing findings in literature, in particular; for the innate immune system in bovine research[65–68]. We also identified enrichment of chemical stimulus involved in sensory perception of smell; we hypothesise that this finding is due to the high number of G-protein coupled receptors assigned this GO annotation and that these receptors may have an alternative function in different tissues, currently as of yet unannotated. The identification of a high number of differentially methylated immune related genes between healthy and BRD calves in this study and the observed overlap of genes with those previously identified by GWAS, transcriptomic and eQTL studies reinforce the biological importance of these candidate genes and pathways. Based on the findings here, regulation of key genes such as *DNAI1, CDCP2* and *ECE1* may have roles in BRD resistance. In particular; due to the structural function in respiratory cilia, roles in vasoconstriction and existing implication in pulmonary fibrosis, *DNAI1* and *ECE1* respectively, are of great interest. Due to its role in complement activation and the existing GWAS and trans-eQTL evidence of the involvement of the complement cascade in BRD, we also consider *CDCP2* a key gene. Although this was a pilot study, it has provided useful insights in BRD resistance in calves confirming that this is a multifactorial trait with differential regulation in genes in immune, inflammatory and pulmonary vasculature pathways among others involved. Our results are

novel and warrant further investigation with a larger sample size to validate the key candidate genes reported here and assess applicability to other cattle breeds as well as provide further insights on calves' resistance to BRD as well as in age related differences. Future studies may also address integration of these results into selective breeding programmes aiming to enhance animal fitness and health.

## Methods
### Ethical approval

This study was performed under the ethical approval of the University of Liverpool Research Ethics Committee (VREC927). Procedures regulated by the Animals Scientific Procedures Act were operated under Home Office License (P191F589B).

### Study Population

This study prospectively enrolled calves at one commercial dairy farm in North Wales. The farm was chosen due to its proximity to the University of Liverpool, and their willingness to participate. Calves were enrolled within one week of birth and were eligible for enrolment if Holstein, female, and were 1 to 7 days old when the weekly visit occurred. A full clinical examination was conducted at <1 week of age ("neo-natal"), 6 weeks of age ("pre-weaning") and 15 weeks of age ("post-weaning"). Calves were housed individually in pens for the first two weeks of life in a farm building dedicated to housing neonatal calves. Thereafter two to three calves were grouped together in pens which had an outside feeding area and a covered deep straw bedded calf hutch. Calves were fed 4 L of colostrum in their first 4-6 h of life using a bucket and teat system, and then placed onto a commercially available milk replacer. Calf starter (concentrates) was available to all calves once moved into grouped housing.

Blood samples were also collected from four adult healthy cows (Holstein, female, 5-6 years old) raised on the same farm. These animals

were sampled as part of a routine herd health screening of freshly calved cows and were clinically healthy at the time of sampling.

## Data Collection

At enrolment (within one week of birth), a blood sample was collected. A full clinical exam was performed by a qualified veterinarian. Animals were also assessed using the Wisconsin scoring system[46] and a composite health scoring system. This scoring system devised by the researchers and is described in supplementary Table S5. Briefly, calves were given clinical examinations at weeks one, five and eight of the study and the measures listed in the table assessed along with record of any discharges associated with eyes and nose, ear position, coughing frequency and whether scour present. Once enrolled, calves were assessed weekly using the Wisconsin scoring system (scores taken for discharges associate with eyes and nose, ear position, coughing frequency and whether scour present) for the presence of bovine respiratory disease. At one, five, and eight weeks, calves underwent a full clinical examination and the Wisconsin scoring system and composite scoring system was used to assess health status. At eight weeks old, in addition to those measures already taken, thoracic ultrasound scanning[85] was also performed to assess respiratory health. Briefly, lungs were examined via ultrasonography and scored on a 0-5 system[85]. Score zero would be considered normal, score one indicates some damage but would not be considered significant. Score two and upwards would be considered significant lung damage. Score two is indicative of a patchy pneumonia/lung damage, whilst score three indicates significant lung damage and subsequent consolidation of a whole lung lobe. Scores four and five are as score three but indicate how many lung lobes are severely affected, such that score four affects two lung lobes and score five affects three or more lung lobes.

The three calves designated healthy control animals were calves (1–3) with no observed incidence of respiratory disease over the study period. The other three calves (10–12) were diagnosed with BRD based on the above-described assessment.

## RRBS sequencing

DNA was extracted from whole blood samples from six calves (three healthy and three diseased) and four adult cows using the DNEasy Kit (Qiagen, Hilden, Germany) according to the manufacturer instructions.

RRBS library preparation and Sequencing (paired-end 50 bp reads on a NovaSeq 6000 platform) was outsourced to Diagenode (Diagenode SA, Rue Bois Saint-Jean, Belgium, https://www.diagenode.com/en) using Illumina technology.

## RRBS sequencing analysis

Initially, FastQC (Andrews, 2010) version 0.11.8 was used for quality control of sequencing reads. Then, the software Trim Galore! Version 0.4.1 (Krueger, 2015) was used to remove sequencing adapters. Subsequently, Bismark version 0.20.0 (Krueger and Andrews, 2011), a specialised tool for mapping bisulfite-treated reads, was used to align reads to the reference genome (bostau8). Bismark requires that the reference genome, in this case bostau8 (https://www.ncbi.nlm.nih.gov/datasets/genome/GCF_000003055.5/), first undergoes in silico bisulfite conversion and transformation of the genome into forward (C - > T) and reverse strand (G - > A).

Reads producing a unique best hit to one of the bisulfite genomes were compared to the non-bisulfite converted genome to identify cytosine contexts (CpG, CHG or CHH – where H is A, C or T). The modules of Bismark version 0.20.0 (Krueger and Andrews, 2011): "cytosine2coverage" and "bismark_methylation_extractor", were used to infer all cytosine methylation states, their context and to calculate percentage methylation. Only CpGs present in all samples were retained for further analysis.

DNA methylation analyses were conducted using methylKit version 1.20.0.[86]. The cytosine2coverage files output from Bismark were used as the methylation call files to create a methylRaw object. Descriptive statistics were calculated, and samples filtered based on read coverage (bases having lower coverage than 10 discarded and bases having higher coverage than

99.9% discarded). Samples were then merged by case/control (diseased/healthy and calf/cow) and correlation between them calculated before identification of differential methylation. Differential methylation in 1000 bp regions using 500 bp steps was identified using Chi-squared test with Sliding Linear Model (SLIM) correction for multiple testing. Differential methylation was determined based on a ≥25% difference in methylation and a statistically significant $p$ value of ≤0.05 and q ≤ 0.01 (q adjusted $p$ value after FDR correction for multiple testing) between sample groups[86–88]. Further, differential methylation at base level resolution was identified, also based on a 25% difference in methylation and a statistically significant $p$ value of ≤0.05 and q ≤ 0.01 between sample groups.

Differential methylation analyses, as described above, were conducted between healthy adult cows and healthy calves as well as healthy calves and those diagnosed with BRD (diseased).

## Functional enrichment and protein interaction network analyses

Genes with associated differentially methylated bases within 50 kb of the transcription start site (TSS) were analysed for gene ontology (GO) term over and under representation to increase understanding of biological pathways and functions that are differentially methylated between healthy cows and calves as well as healthy and diseased calves.

GO term over and under representation was assessed with DAVID functional enrichment and functional annotation clustering analysis. The *Bos taurus* background information is available in DAVID and was used for the analysis. The enrichment score (ES) of the DAVID package is a modified Fisher exact $p$ value calculated by the software, with higher ES reflecting more enriched clusters. An ES greater than 1 means that the functional category is overrepresented. Moreover, for the DAVID functional enrichment analysis a statistical correction for multiple testing was applied with FDR adjusted $p$-value (or q-value) ≤0.05 set as the statistically significant threshold identified.

Over and under representation of terms was also identified using PANTHER (v17.0)[49] overrepresentation test. Pathway enrichment was also assessed using Reactome v. 77[50] against the Bos Taurus genome (Ensembl release 108). Both PANTHER overrepresentation test and Reactome pathway enrichment used FDR calculations to correct for multiple comparisons, therefore, statistical significance was determined for both based on FDR adjusted $p$-value (or q-value) ≤0.05, the default correction for these tools.

Protein interaction networks for differentially methylated regions within 1 kb of a TSS in each comparison were analysed using STRING (v 11.5)[55] with parameters: minimum required interaction score - highest confidence (0.900), no second shell interactions, network edges - confidence, disconnected nodes hidden, k-means clustering of interactions into 3 clusters.

## Reporting summary

Further information on research design is available in the Nature Portfolio Reporting Summary linked to this article.

## Data availability

Raw data and methylation call files are available at ENA under the project accession PRJEB71365. Source data are available in Supplementary Data 1.

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

## Acknowledgements

We acknowledge the financial support provided by the Biotechnology and Biological Sciences Research Council (BB/S002960/1, BB/S003614/1, BB/S002944/1). The authors express gratitude to the farm staff and herd managers for their assistance and cooperation without which this study would not have been possible.

## Author contributions

A.P., G.O., and G.B.: conceived and designed the study and secured funding. B.G. and G.O. collected the samples and performed the phenotyping. E.A. performed the RRBS analysis and wrote the manuscript with input from all authors. A.P. extracted DNA and supervised the RRBS analysis. D.X. provided expertise on enrichment analyses and D.W. expertise on BRD immunological responses. All authors and K.P. contributed to the interpretation of the results and assisted in revising the manuscript.

## Competing interests

The authors declare no competing interests.
