## [Peer Review File · Communications Biology]

Reviewers' comments:

Reviewer #1 (Remarks to the Author):

This manuscript identifies methylation markers for age and BRD in dairy cattle. The authors selected 3 "healthy" and 3 "affected" calves based on health scores taken at 1, 5, and 8 weeks of age. Additionally, 2 adult cows were used to identify age related methylation. Genes showing differential methylation (increased or decreased) are presented as are pathways. The manuscript provides new and useful information, however, it may have been improved by using a larger number of animals and including breeds other than just Holsteins. Specific concerns are listed below.

1 L34 suggest revising the term "a lot"

2 L44 suggest revising to read "Only recently in cattle have epigenetic clocks..."

3 L64-74 what about vaccination for BRD? How does this contribute to health and productivity?

4 L84 "improving genetic resistance" to what in particular?

5 L93 Results: the difficulty in reading a manuscript formatted in this way is that the reader does not know things listed in the materials and methods section such as the age of the calves. It would be helpful to include some information as you describe the results.

6 L107/Table Please differentiate between the healthy and affected animals.

7 L132-133 The authors list differentially methylated genes (DAVID) yet then state that after Bonferroni correction, no terms were significant. Please include here and in the Materials and Methods section why Bonferroni correction is used, and if it is important, perhaps that is the data that should be shown? Additionally, in this section and L283-306 and similar sections, the results would be easier to read if they were in tables similar to Table 2. Please reformat.

8 L138-142 this section is difficult to understand

9 Figures 3 and 5, recommend using a different colorway - it is difficult to differentiate between the red and brown.

10 Figures 4 and 6 please include color code to differentiate between these groups

11 L220 delete "bases"

12 L233-234, 259-260 - Bonferroni as stated above

13 L456 consider bolding Conclusions

14 L516 also list 10-12 to describe the BRD calves

15 L521 include the address of Qiagen

16 L524 include the address of Diagenode

17 L532-533 please include the source - currently states (source required)

18 L535 revise "genomewere" to "genomes were"

19 Include a paragraph in Materials and Methods on statistical analysis including Bonferroni correction

20 L560-561 revise "For the twin comparison..." to "Due to the high number of identified genes in the twin comparison, only those..."

Reviewer #2 (Remarks to the Author):

The authors of the present work used RRBS (Reduced Representation Bisulfite Sequencing) to study methylation differences between healthy cows and calves to identify potential markers of genetic age. They also examined differences in methylation between healthy and BRD (Bovine Respiratory Disease)-affected animals. Their major claims consist in the identification of differentially methylated regions between calves and cows with the enrichment in genes responsible for developmental pathways such as cell fate commitment and tissue morphogenesis. Between healthy and BRD-affected calves the functional analysis pointed to enrichment in genes responsible for immune and vasculature regulatory pathways and the candidate genes for the BRD susceptibility involved in complement cascade regulation, vasoconstriction, and respiratory cilia structure and function were identified. The authors' results are novel, and convincing, and will be interesting to others in the community and the wider field. The authors' results may also significantly contribute to the genetic improvement of dairy cattle. I believe that the statistical methods used were appropriate and valid and based on the provided details, the work is fully reproducible.

I recommend the article for publication, I have no fundamental reservations, just a few comments concerning the formal aspect:

- line 95: "Table 1" could be in brackets as it is in the further text
- line 220: "1541 regions bases were hypermethylated"?
- line 357-358: AKT2 is involved in macrophage polarization (ablation results in M2 macrophages involved in wound healing)? I do not understand this expression, is not there something missing?
- line 365: the term "promoter" or "promotor"?
- line 373: is the term "organismal" correct?
- line 442: should not be - CUB domain-containing protein 1?
- line 446: the abbreviation of gene DNAI1 could be explained as others
- line 535: genomewere

Reviewer #3 (Remarks to the Author):

The authors report differences in methylation between healthy dairy calves, calves afflicted with BRD, and mature healthy cows. These methylation differences were then used to infer underlying genes related to BRD susceptibility. The study idea is well-founded given, as stated by the authors, BRD susceptibility has been shown to be heritable and methylation patterns are impacted by age. The general premise that disease status impacts methylation is certainly of interest to a broad audience and lends to more expansive work to understand how these changes impact traits measured much later in life. Being able to demonstrate that methylation changes, due to early life health events, are associated with later in life performance (fertility, milk yield, disease

susceptibility) would bolster a study such as the current one.

However, my primary concern is the number of animals used in the current study. Having only 3 cases and 3 controls is very limiting. There are many environmental factors that could impact methylation and a limited sample size cannot account for these. Moreover, using only 2 cows as the "control" for age related methylation is not sufficient. These two adults could differ in life experiences and prior health history and alone cannot sufficiently control for differences due to age. I believe these shortcomings make inferences valid to a very small group of animals and not the broader population.

Androniki Psifidi
Royal Veterinary College
Hawkshead Lane
North Mymms
Hertfordshire
AL9 7TA

Dear Reviewers,

We would like to thank you for your positive assessment and the time and effort put in to reviewing the previous version of our manuscript (COMMSBIO-24-0307) entitled "Identification of methylation markers for age and Bovine Respiratory Disease in Holstein-Frisian cattle".

We have now revised the manuscript and taken into account all the comments and suggestions, detailed in our point by point response to reviewers below. Specifically; brief introductions have been added to each results subsection to aid reader understanding, colour schemes of plots have been changed to improve differentiation of points and the sections identified as difficult to understand have been reformatted and edited to improve clarity. More detail has also been added to the description of the statistical analysis to ensure reader understanding. Significantly, we also dedicated additional resources to increase the sample size of our adult cow population from two to four and repeated the age related RRBS comparison.

We therefore believe that our thorough revision and additional work will satisfy the comments made during review and will permit publication of our findings which are valid and valuable to the scientific community.

We have no conflicts of interest to disclose.

Thank you for your review of this manuscript.

Sincerely,

Dr Androniki Psifidi PhD, DVM, PGCVetEd, FHEA, AUA
Senior Lecturer in Veterinary Clinical Genetics

Reviewer	Comment	Response
R1	This manuscript identifies methylation markers for age and BRD in dairy cattle. The authors selected 3 "healthy" and 3 "affected" calves based on health scores taken at 1, 5, and 8 weeks of age. Additionally, 2 adult cows were used to identify age related methylation. Genes showing differential methylation (increased or decreased) are presented as are pathways. The manuscript provides new and useful information, however, it may have been improved by using a larger number of animals and including breeds other than just Holsteins. Specific concerns are listed below.	We would like to thank the reviewer for their positive assessment and their time reviewing our work and providing comments. Regarding sample size please see relevant reply above. Briefly, acknowledging the reviewer 's concern regarding sample size we committed the effort and the resources to increase our sample size for the age comparison (raising the number of adult cows from 2 to 4), repeat the age comparison and update the manuscript's results and discussion. Please see below detailed responses to specific comments.
R1	1 L34 suggest revising the term "a lot"	Done. Edited in the ext "Although there are relevant studies in humans and mice, there are limited studies in cattle." In lines 36 of the revised manuscript
R1	2 L44 suggest revising to read "Only recently in cattle have epigenetic clocks..."	Done. Edited in the text: "Only recently have epigenetic clocks been constructed in cattle predict chronological age...." In lines 44-46 of the revised manuscript
R1	3 L64-74 what about vaccination for BRD? How does this contribute to health and productivity?	Done. Relevant text to discuss vaccination for BRD added in lines 74-79 of the revised manuscript.
R1	4 L84 "improving genetic resistance" to what in particular?	Done. Improving genetic resistance to BRD infection, corrected in the text in line 91 of the revised manuscript

R1	5 L93 Results: the difficulty in ready a manuscript formatted in this way is that the reader does not know things listed in the materials and methods section such as the age of the calves. It would be helpful to include some information as you describe the results.	Done. Introductory sentences have been added to the start of results subsections to aid understanding prior to reading materials and methods. For example, in lines 102-109, 123-124 and 225-226 of the revised manuscript
R1	6 L107/Table Please differentiate between the healthy and affected animals.	Done. Relevant clarification added in lines 119-120 in the Table 1 title.
R1	7 L132-133 The authors list differentially methylated genes (DAVID) yet then state that after Bonferroni correction, no terms were significant. Please include here and in the Materials and Methods section why Bonferroni correction is used, and if it is important, perhaps that is the data that should be shown? Additionally, in this section and L283-306 and similar sections, the results would be easier to read if they were in tables similar to Table 2. Please reformat.	We now consider FDR correction for multiple testing across all analyses in the manuscript, since it is a robust, strict and widely-accepted method. All the algorithms we have used for our analyses are using automatically FDR correction for multiple testing. For the DAVID functional enrichment analysis the software also produced results with the Benjamin and Bonferroni correction, and they were in full agreement with FDR. We have clarified this further in the revised manuscript. We have also added relevant text throughout the Materials and Methods section to describe the significance thresholds we have used. To ease reading as suggested by the reviewer some results have been presented in the format of Tables. Specifically, two more Table (Table 3 and 5) have been added in the revised manuscript.
R1	8 L138-142 this section is difficult to understand	Done. The structure and wording of this section has been revised to improve understanding. Please see revised text in lines 150-158 of the revised manuscript.

R1	9 Figures 3 and 5, recommend using a different colorway - it is difficult to differentiate between the red and brown.	Done. Colour palette for plots was updated to increase clarity.
R1	10 Figures 4 and 6 please include color code to differentiate between these groups	Done. Colour code has been added and relevant clarification in the figure legends "Colour indicates the kmeans clustering of interactions."
R1	11 L220 delete "bases"	Done. Corrected in the text, line 228 of the revised manuscript
R1	12 L233-234, 259-260 - Bonferroni as stated above	Please see response above
R1	13 L456 consider bolding Conclusions	Done. Conclusions bolded, line 467 of the revised manuscript
R1	14 L516 also list 10-12 to describe the BRD calves	Added in the text, line 497-508
R1	15 L521 include the address of Qiagen	Done. Added in the text, line 538 of the revised manuscript
R1	16 L524 include the address of Diagenode	Done. Added in the text, line 541 of the revised manuscript
R1	17 L532-533 please include the source - currently states (source required)	Done. Added in the text, line 549 of the revised manuscript
R1	18 L535 revise "genomewere" to "genomes were"	Done. Corrected in the text, line 552 of the revised manuscript
R1	19 Include a paragraph in Materials and Methods on statistical analysis including Bonferroni correction	Details added in the text to aid clarity of statistical analysis, lines 578-585
R1	20 L560-561 revise "For the twin comparison..." to "Due to the high number of identified genes in the twin comparison, only those..."	The twin comparison was removed from the initial paper submission. Relevant text has now removed.

R2	The authors of the present work used RRBS (Reduced Representation Bisulfite Sequencing) to study methylation differences between healthy cows and calves to identify potential markers of genetic age. They also examined differences in methylation between healthy and BRD (Bovine Respiratory Disease)-affected animals. Their major claims consist in the identification of differentially methylated regions between calves and cows with the enrichment in genes responsible for developmental pathways such as cell fate commitment and tissue morphogenesis. Between healthy and BRD-affected calves the functional analysis pointed to enrichment in genes responsible for immune and vasculature regulatory pathways and the candidate genes for the BRD susceptibility involved in complement cascade regulation, vasoconstriction, and respiratory cilia structure and function were identified. The authors' results are novel, and convincing, and will be interesting to others in the community and the wider field. The authors' results may also significantly contribute to the genetic improvement of dairy cattle. I believe that the statistical methods used were appropriate and valid and based on the provided details, the work is fully reproducible. I recommend the article for publication, I have no fundamental reservations, just a few comments concerning the formal aspect:	We would like to thank the reviewer for their endorsement of our work, their comments as well as their time in reviewing our work. Please see below detailed response addressing all the comments.
R2	- line 95: "Table 1" could be in brackets as it is in the further text	Done. Added in the text, line 110 of the revised manuscript.

R2	- line 220: "1541 regions bases were hypermethylated"?	Done. Corrected in the text ('bases' was a typo and removed), line 228 of the revised manuscript.
R2	- line 357-358: AKT2 is involved in macrophage polarization (ablation results in M2 macrophages involved in wound healing)? I do not understand this expression, is not there something missing?	Done. Sentence was rewritten for clarity: "It is, however, important to also note the role of AKT2 in the immune system. AKT2 is involved in macrophage polarization, regulation of the functions of dendritic cells and proliferation of T regulatory cells and AKT2 ablation has been shown to results in macrophage polarisation to M2 macrophages, involved in wound healing.", in lines 364-367 of the revised manuscript.
R2	- line 365: the term "promoter" or "promotor"?	Done. Corrected throughout the revised manuscript to promoter
R2	- line 373: is the term "organismal" correct?	Done. Corrected in the text, line 382 of the revised manuscript.
R2	- line 442: should not be - CUB domain-containing protein 1?	Done. Corrected in the text, line 452 of the revised manuscript.
R2	- line 446:the abbreviation of gene DNAI1 could be explained as others	Done. Definition added in the text, line 426 of the revised manuscript.
R2	- line 535: genomewere	Done. Corrected in the text, line 456-457 of the revised manuscript.
R3	The authors report differences in methylation between healthy dairy calves, calves afflicted with BRD, and mature healthy cows. These methylation differences were then used to infer underlying genes related to BRD susceptibility. The study idea is well-founded given, as stated by the authors, BRD susceptibility has been shown to be heritable and methylation patterns are impacted by age. The general premise that disease status impacts methylation is certainly of interest to a broad	We would like to thank the reviewer for their positive comments and assessment as well as for their rigorous review.

	audience and lends to more expansive work to understand how these changes impact traits measured much later in life. Being able to demonstrate that methylation changes, due to early life health events, are associated with later in life performance (fertility, milk yield, disease susceptibility) would bolster a study such as the current one.	
R3	However, my primary concern in the number of animals used in the current study. Having only 3 cases and 3 controls is very limiting. There are many environmental factors that could impact methylation and a limited sample size cannot account for these. Moreover, using only 2 cows as the "control" for age related methylation is not sufficient. These two adults could differ in life experiences and prior health history and alone cannot sufficiently control for differences due to age. I believe these shortcomings make inferences valid to a very small group of animals and not the broader population.	We acknowledge reviewer's concern regarding sample size and we committed the effort and the resources to increase our sample size for the age comparison (raising the number of adult cows from 2 to 4). We have updated the results and discussion section in the revised manuscript to reflect this new analysis. Some of the additional findings from the new analysis of age comparisons have been also added to the supplementary material tables. With respect to the BRD comparison acknowledging the importance of the environment we have attempted to mitigate the influence of different environmental factors as much as possible within a 'real world' population by ensuring animals were all from the same farm and under the same management practices as well as ensuring all those included were genetically related (at least half siblings). Moreover, we have in the abstract and discussion made clear the sample size and toned down wording to reflect the small sample size and the need for further studies.

Reviewers' comments:

Reviewer #1 (Remarks to the Author):

The authors have addressed all comments and the manuscript should be accepted for publication.

Reviewer #3 (Remarks to the Author):

I appreciate the authors efforts in revising their manuscript, particularly in increasing the sample size.

Despite these revisions, my concerns relative to the sample size remain. It is my belief that 3 "healthy" calves and 3 "diseased" calves is simply too small. Moreover, the differences between "healthy" and "diseased" based on Table 1 do not seem to be large. Example: There appear to be minimal differences between calf 3 and calf 10 for the Wisconsin scores and the lung scores. It is not just the number of calves used that is of concern, it is the size of the cohort used to select calves which limited the degree to which "cases" and "controls" differed.

Given the authors, as well as others, have described the impact of age on methylation patterns, why did the authors not select mature cows with more similar ages? Admittedly 1 year (5-6 years) does not seem large, but less variation here would seem helpful. Alternately, variation in cow age could be introduced if there were multiple females from each age class (e.g., 2, 3, 4, 5, 6 yrs. of age).

Line 63: Up to 12% of what?

Line 116: I would delete "full siblings" given it is possible to be a full-sib but not a twin and adding this after you have stated they are a twin adds no new information.

Reviewer	Comment	Response
R1	The authors have addressed all comments and the manuscript should be accepted for publication.	We would like to thank the reviewer for their endorsement of our manuscript for publication.
R3	Despite these revisions, my concerns relative to the sample size remain. It is my belief that 3 "healthy" calves and 3 "diseased" calves in simply too small. Moreover, the differences between "healthy" and "diseased" based on Table 1 do not seem to be large. Example: There appear to be minimal differences between calf 3 and calf 10 for the Wisconsin scores and the lung scores. It is not just the number of calves used that is of concern, it is the size of the cohort used to select calves which limited the degree to which "cases" and "controls" differed. Given the authors, as well as others, have described the impact of age on methylation patterns, why did the authors not select mature cows with more similar ages? Admittedly 1 year (5-6 years) does not seem large, but less variation here would seem helpful. Alternately, variation in cow age could be introduced if there were multiple females from each age class (e.g., 2, 3, 4, 5, 6 yrs. of age).	We acknowledge reviewer's concern regarding sample size and other limitations of this study, therefore, we have included a paragraph discussing in detail these limitation And further we have toned down the title, abstract and conclusions making clear the sample size used and presenting the results as o fa pilot study. Specifically; the title has been altered to: "Identification of DNA methylation markers for age and Bovine Respiratory Disease in dairy cattle: A pilot study based on Reduced Representation Bisulfite Sequencing". In lines 22-23 of the abstract "We have performed a pilot study to investigate the epigenetic profile of BRD susceptibility in six calves....".

		We have included a limitations paragraph lines 471-477 In line 496-497 of the conclusion “Although this was a pilot study, it has provided useful insights in BRD resistance ...” In line 499-500 of the conclusion “Our results are novel and warrant further investigation with a larger sample size to validate....” We feel the age of the adult cows (5-6 years old) is a small difference between individuals and accurately represents fully mature animals and is sufficiently different to the calves for an age comparison.
R3	Line 63: Up to 12% of what?	“up to 12% of early calf losses”, corrected in the text, line 65 in the revised manuscript.

R3	Line 116: I would delete "full siblings" given it is possible to be a full-sib but not a twin and adding this after you have stated they are a twin adds no new information.	Corrected in the text "with calves 3 and 12 being twins with different BRD diagnoses" lines 118-119 of the revised manuscript
----	--	---